# Sexual Identity–Behavior Discordant Heterosexuals in Britain: Findings from the National Survey of Sexual Attitudes and Lifestyle 2010–2012 (Natsal-3)

**Maria Calatrava** [1,2,3,*] **, D. Paul Sullins** [4,5] **and Steph James** [6]

1    Institute Culture and Society, University of Navarra, 31008 Pamplona, Spain
2    IdiSNA, Navarra Institute for Health Research, 31008 Pamplona, Spain
3    School of Education and Psychology, University of Navarra, 31008 Pamplona, Spain
4    Department of Sociology, The Catholic University of America, Washington, DC 20064, USA; sullins@cua.edu
5    Ruth Institute, Lake Charles, LA 70601, USA
6    Independent Researcher, London W1G 8AX, UK; steph1@polopostal.eu
*    Correspondence: mcalatrava@unav.es; Tel.: +34-948-425-600

**Abstract:** This study delves into heterosexual identity–behavior discordance, exploring the interaction between sexual identification and behavior in the UK. Analyzing representative 2010 data from the British population (N = 12,472), this research categorizes participants into different groups: nonheterosexual, concordant heterosexual, and three discordant heterosexual categories (closeted, experimenters, and desisters). These groups were compared in terms of sociodemographics, attitudes toward sexuality, risk behaviors, and health indicators. Discordance was associated with permissive social attitudes, including behaviors such as adultery and one-night stands, and with increased substance abuse and risky sexual behaviors, especially among the closeted. Surprisingly, the physical and mental health of discordant heterosexuals was similar to concordant heterosexuals, in contrast to the poorer health of nonheterosexuals. Due to the significant variations in lifestyles and health indicators among different groups, this study highlights the importance of providing targeted services and interventions.

**Keywords:** sexual orientation; sexual identity; discordance; sexual minority; risk behaviors; mental health

## 1. Introduction

Discordance among the dimensions of sexual orientation presents both theoretical and practical challenges for the study of sexual minorities [1]. Theoretically, scientists have conceived of human sexuality as "oriented" to more or less consistent behavior among three distinct experiences: sexual attraction (what sex persons are sexually attracted to), sexual behavior (what sex persons have sex with, e.g., same-sex or opposite-sex sex partners), and sexual identity (what persons call themselves, e.g., "gay/homosexual", "bisexual", or "straight/heterosexual") [2]. On this view, human sexuality is socially organized on a spectrum from heterosexual-identified persons who are attracted to and have sex with members of the opposite sex on one end to homosexual-identified persons who are attracted to and have sex with members of the same sex on the other end. The widely used Kinsey Scale, developed 75 years ago [3], embeds this assumption of zero-sum sexual orientation by requiring that respondents who indicated greater same-sex attraction thereby simultaneously indicated lower opposite-sex attraction on a one-to-seven scale ranging from "heterosexual" at one end to "homosexual" at the other. The expectation is that persons can be classified more or less coherently as homosexual, heterosexual, or something in between. This conception may stem in turn from the heteronormative idea that same-sex sexuality is but a mirror image of opposite-sex sexuality. Discordance directly challenges such a classification scheme [4,5].

Practically, many studies of sexual minority populations, and clinical intake processes, have not measured all three dimensions but assumed that sexual orientation can be reliably inferred from only one or two dimensions. Studies of "men who have sex with men" (MSM) or "women who have sex with women" (WSW) often ignore the other two non-behavioral dimensions [6]. Public health surveys frequently infer sexual orientation from identification alone [7,8]. Studies that classify persons as "bisexual" based on lifetime sexual experience with both men and women likely inadvertently include currently discordant persons in this category [9]. Discordant persons, moreover, may not choose any of the three dimensions in favor of no response or "something else" [10]. Such measurement does not accurately capture discordant sexual minorities, not to mention those questioning or who think in more diverse ways about their sexual orientation [11], thereby underestimating the size and diversity of the sexual minority population and potentially excluding the very group of persons who may be in the most need of health interventions.

The recognition that individuals may experience discordance among two or more of the dimensions of sexual orientation, meaning that they may not be aligned in the same direction, calls for new classification and research. Studies have shown that the three dimensions exhibit imperfect overlap and inconsistent predictability with one another [12,13] and that each dimension separately predicts differing health disparities [5]. Therefore, measuring only one dimension without the inclusion or consideration of the others may lead to simplicities and reductionism in research and the misclassification of treatment needs in clinical settings. The U.S. National Academy of Science recently noted that sexual orientation measurement inaccuracies "are not purely academic: they can have severe consequences for sexual and gender minorities in health care and other areas in which measures of sex/gender and sexual orientation are often used for determining appropriate and necessary care" [14].

Research on sexually discordant persons may help improve both theory and practice and improve understanding of this doubly marginal population. A repeated finding, for example, is that rates of incongruity are higher among nonheterosexuals. Laumann et al. [13] were among the first to report that while these three dimensions were highly congruent for the heterosexual majority, this was not the case for the nonheterosexual minority. In their survey of a representative sample of the American population, they found that of persons who experienced same-sex orientation on any one of the three dimensions, only 24% of males and 15% of females experienced it on all three dimensions [13,15]. Similarly, Geary et al. found in a British survey conducted in 2010–2012 that only 26% of nonheterosexual men and 14% of nonheterosexual women reported same-sex orientation on all three dimensions [12].

Recent research has also shed light on the fluidity of these three dimensions, emphasizing the dynamic nature of sexual orientation across a lifespan and its bidirectionality over time [16–19]. This statement appears particularly relevant to individuals who have ever encountered feelings of same-sex attraction. According to a study of sexual attraction change in four US longitudinal studies, between 26% and 64% of people with same-sex attraction reported changing sexual attraction over time; between one-half and two-thirds of those who changed switched to heterosexuality, while a very small proportion (between 1% and 12%) switched from exclusive opposite-sex attraction to same-sex attraction [1]. Another study conducted in a large sample in the United Kingdom highlighted that the rate of sexual identity fluidity over a 6-year period is relatively low among those who previously self-identified as heterosexual (3.3%), higher among those who self-identified as gay/lesbian (16.1%), and particularly high among those with bisexual (56.8%) and other sexual identities (85.4%) [20]. In addition, the fluidity of attraction is likely to be accompanied by increased instability of sexual behavior and partner sex. This is particularly evident for bisexual persons [21], but it is not exclusive to them [16]. Some authors even consider that different types of attraction and behaviors seem to converge simultaneously in some people, mainly nonheterosexuals, and in a minority of heterosexuals referred to

as discordant. These studies highlight that sexual attraction and behavior may be closely related to each other, and a change in one could lead to a change in the other.

By the same token, sexual identity stands somewhat over against both attraction and behavior. It is common to find greater discrepancies between identity and attraction and/or identity and behavior than between attraction and behavior [12,13]. Studies have also suggested that sexual identity could respond to factors other than attraction and behavior. As a developmental process of intentional self-identification that may be more susceptible to social influences and maturation processes, the relationship of sexual identification to fluidity in attraction and/or behavior is of special interest for understanding the establishment and stability of sexual minority identity. Therefore, the analysis of the mismatch between identity and the history of either attraction or behavior is of special interest.

### 1.1. Heterosexual Identity–Behavior Discordance (IBD)

While data constraints preclude the study of attraction history—no population survey (to our knowledge) has asked about past sexual attraction—with the recent appearance of national surveys that capture sexual partner histories by sex of partner, it has become possible to examine developmental questions of identity–behavior discordance (IBD). Due to the small size of the nonheterosexual population, such data have mostly supported inferences about IBD in the heterosexual population, that is, persons who identify as heterosexual yet engage, or have engaged in the past, in same-sex encounters. In the last decade, several studies have reported the presence and size of this population. In the United States, the percentage of heterosexual-identified persons reporting a same-sex sexual partner was 10.2% in females and 2.6% in males. A very similar prevalence was obtained in Australia (10.9% in females and 3.7% in males). The lowest prevalence of behavioral discordance was found in Canada, where 2.7% of females and 0.7% of males, respectively, reported it [22]. In Britain, the Natsal-3 population study reported that 5.5% of men and 6.1% of women had had same-sex partners and of these, the majority self-labeled as heterosexual [12]. Notably, studies have found that sexual IBD is especially prevalent among young adult women [22–24]. Studies have also revealed that individuals classified as IBD reported lower levels of physical health and psychological well-being compared to concordant heterosexuals [23] and engaged in negative behaviors such as binge drinking [24], revealing the potential negative implications for general health associated with discordance. Apart from sex, however, few studies explored associations between discordance and various demographic factors [22] and only limited research, as already noted, has examined the related development and health disparities of the heterosexual population who reported IBD.

The reasons why a person becomes identity–behavior discordant may vary. For example, some authors speculated that heterosexual women are more likely than men to be discordant, possibly as a result of a more fluid sexuality that makes them more susceptible to same-sex behavior [16]. Other authors argue that younger generations may show a greater propensity to engage in discordant sexual practices as a way of affirming their lack of interest in binary categories of opposite-sex or same-sex partners [19,25,26]. This behavior may be the result of a cultural trend toward a greater acceptability of same-sex romantic behaviors today and the social invitation to self-discover one's sexual orientation. Additionally, alternative viewpoints propose that sexual discordance might be attributable to psychological factors, such as the presence of internalized homophobia. This approach suggests that nonheterosexual persons may conceal their authentic sexual orientation within a heteronormative environment in order to avoid stigma [22,27,28]. As such, there could be multiple forms of discordance and probably different pathways to reach it, so classifying all individuals as identity–behavior discordant does not imply that they are all similar.

Among the conceptual frameworks identified to explain disparities in health issues between heterosexuals and nonheterosexuals, the minority stress theory is probably the most common. This model focuses specifically on sexual-orientation-related stressors. It

identifies distal stressors (e.g., social stigma, adverse childhood experiences) and proximal stressors (e.g., internalized homophobia, concealment of sexual orientation) that can lead to adverse health outcomes [29]. Research on bisexual persons, who also suffer higher mental health disparities than other sexual minorities, has proposed that they may face added stigma due to "bisexual invisibility/erasure, experiences of bisexual-specific discrimination, biphobia in the gay community, and lack of support for bisexual sexuality" [30]. IBD persons may face similar elevated stigma, including IBD phobia in the gay community. A second model widely recognized in the literature is the socioecological model. This framework emphasizes the role of social factors, such as socioeconomic status, education, employment, and access to healthcare, in shaping health outcomes. It suggests that sexual minority individuals may face distinct social disadvantages and inequities that contribute to disparities in health [31]. Finally, recent genetic studies pointed out that nonheterosexual persons share a genetic predisposition toward risky behaviors. A large genome-wide-association study (GWAS) by Ganna et al. [32] found that nonheterosexual sexual behavior was related to a genetic propensity for risk-taking behaviors related to sexual health. This study confirmed the findings of an earlier, smaller GWAS that had reported the same result [33]. These frameworks are not mutually exclusive and often overlap, creating intricate intersections. All of them might offer diverse perspectives to researchers and practitioners in an attempt to explain the health disparities experienced by sexual minority individuals.

*1.2. Objective*

Currently, the main constraint to a better understanding of IBD sexuality, and what this may contribute to both the theoretical interpretation of sexual orientation and to the health needs of the IBD population, is an extreme lack of representative parametric information about this population. The objective of the present study is modest but important: to provide an initial demographic and attitudinal profile of IBD heterosexuals which can form a basis for further theoretical and practical work regarding this population.

Analyzing representative data of the British population in 2010, we identify three unique sub-groups by sexual partner history that exhaustively comprise the IBD population which we term closeted, experimenters, and desisters. After presenting the prevalence and population characteristics of these groups, we compare and contrast them, also with concordant heterosexuals and self-identified nonheterosexuals, regarding pertinent sociodemographic factors, attitudes toward sexual practices and LGBT rights, risk behaviors, and health status indicators.

Unearthing the unique profiles of these groups, we aim to provide a general picture of IBD heterosexuals that will establish baseline empirical information that may be useful, in a small way, for thinking about the complex relationship between fluid sexual behavior and sexual identity and the diverse nature of sexual orientation. We also hope to provide initial information to help better serve the unique health needs of this often overlooked part of the sexual minority population.

## 2. Materials and Methods

*2.1. Sample*

Comprehensive descriptions of Natsal-3's design and methods have been published elsewhere [34,35], to which we refer the interested reader. Here, we present a brief summary pertinent to the present study.

From September 2010 to August 2012, Natsal-3 interviewed 15,162 household residents aged 16–74 in England, Scotland, and Wales, selected using a stratified, multi-stage cluster-sampling frame designed to be statistically representative of the British population. The contact response rate was 57.7%. Individuals living in communal establishments, such as military barracks, prisons, and boarding schools, were excluded from the sample.

Although over a decade old, recent data (2021) indicate little change in the prevalence of the LGB population since the Natsal-3 study was conducted. In Natsal-3, 2.8% of the British population self-identified as gay, lesbian, or bisexual. The most recent nationally

representative polling of the UK population (2021) showed that 3.1% of the population identified at lesbian, gay, or bisexual [36]. At least in terms of global prevalence, the Natsal-3 data correspond closely to the current distribution of the British population.

*2.2. Instruments and Variables*

2.2.1. Sexual Identity

During the interview, male (female) participants were shown three cards related to the dimensions of sexual orientation, with options associated with random letters of the alphabet, and asked to indicate which letter best represented themselves to the interviewer, who entered the letter into a computer. For "Sexual identity", the respondents were asked "Which of the options on this card best describes how you think of yourself?". The response options, conforming to the guidelines of the Office of National Statistics [37], were "Heterosexual/Straight; Gay/Lesbian; Bisexual; Other". In this study, this variable was recoded in two categories: 0 = heterosexual and 1 = nonheterosexual.

Natsal-3 referred to gender and sex interchangeably and did not distinguish between male/female and man/woman dichotomies. This paper necessarily follows this usage.

2.2.2. IBD Heterosexual Groups

We created a new variable with five categories which refer to each group of analysis.

Individuals who at the time of the survey identified themselves as heterosexual and reported having had only opposite-sex partners in their lifetime were classified as (0) concordant heterosexuals. Conversely, if participants self-identified as heterosexual but reported one or more lifetime same-sex partners, they were classified as discordant heterosexuals. Discordant heterosexuals were further categorized into three groups: experimenters, closeted, and desisters. (1) Closeted discordants consisted of self-identified heterosexuals who reported having continually had one or more same-sex partners, both over the course of their lifetime and in the past year. While identifying as heterosexual, and not gay/lesbian or bisexual, while currently engaging in same-sex sexual relationships is consistent with a degree of identity non-disclosure, it is possible that these individuals reported a different sexual identity in other settings. (2) Desister discordants consisted of self-identified heterosexuals who reported multiple same-sex partners in the past, that is, over their lifetime, but no same-sex partners recently, that is, in the past year. Most persons in this group (62.8%; 95% CI, 53.5–71.2) also had had no same-sex partners but only heterosexual partners in the past five years. (3) Experimenter discordants were like desisters but with the difference that they reported only one past same-sex partner over their lifetime and no same-sex partners in the past year. The classification of closeted, desisters, and experimenters was both exhaustive and mutually exclusive; all discordant heterosexuals could be placed into a single category within the group of discordant heterosexuals. Lastly, those participants who self-identified as (4) nonheterosexual were classified into another group.

Since we were interested in the relationship between sexual identity and sexual behavior, we analyzed only those who reported having had at least one sexual partner in the past five years (N = 12,472). This excluded 2690 cases consisting of those who reported zero sexual partners in the past five years (N = 2148) and nonrespondents (N = 542).

2.2.3. Attitudes toward Sexuality and Pro-LGBT Rights

Three items that measured different opinions toward questionable sexual practices (i.e., adultery, one-night stands, and sex without love) were selected. For each of these three items, a new variable was created in order to compare those participants who agree with "adultery was rarely or never wrong" (versus mostly or always wrong), those who agree "One-night stand rarely or never wrong" (versus always wrong) and those who strongly agree "Sex without love is OK" (versus agree or disagree).

Pro-LGBT attitudes were measured through 4 different items: "Sexual relations between two adult men are right", "Sexual relations between two adult women are right", "Gay men should be able to adopt children", and "Lesbians should be able to adopt chil-

dren". The two first items used a 5-point scale with the categories of 0–4 ranging from always, mostly, sometimes, or rarely wrong and 5 indicating "not wrong at all". For both variables, over half of the respondents indicated "not wrong at all", with the remainder spread across the categories indicating whether the behaviors were considered more or less often wrong. For descriptive purposes, responses to each item were dichotomized as 0 = else (from 1 to 4) and 1 = not wrong at all (5), contrasting those with complete tolerance with all other responses, including those who reported "depends/I don't know". The two adoption items used a Likert scale ranging from 0 = strongly disagree to 5 = strongly agree. In order not to lose statistical power in a multivariate analysis, a single measure of pro-LGBT attitudes was created, obtaining the mean score of the four attitudes. Internal consistency was very good (Cronbach's alpha = 0.90). This variable was dichotomized by the median to create groups with high and low pro-LGBT attitudes.

### 2.2.4. Sexually Transmitted Infection (STI) Risk Behaviors

The linear trend in the number of sex partners (same sex or opposite sex) in the last 5 years was calculated. Perceived STI risk was measured through an item which had 4 possible responses (1 = greatly at risk; 2 = quite a lot; 3 = not very much; 4 = not at all at risk). This variable was dichotomized into two categories (0 = not very much or not at all at risk; 1 = quite a lot or greatly at risk). Having ever paid for sex was measured through a dichotomous variable (0 = no and 1 = yes).

### 2.2.5. Substance Use

Substance use was measured through the following 6 items: (1) ever smoked tobacco; (2) currently smoking tobacco, which had 4 possible answers (1 = nonsmoker, 2 = ex-smoker, 3 = light smoker, and 4 = heavy smoker); (3) number of cigarettes a day (continuous); (4) average alcohol consumption per week, with 4 possible answers (1 = none, 2 = low (women <=14, men <=21 units per week), 3 = moderate (women > 14–35, men > 21–50 units per week), and 4 = high (women > 35, men > 50 units per week)); (5) weekly binge drinking or more often, with 5 possible answers from 1 = never to 5 = daily or almost daily; (6) smoked marijuana in the past year, with two possible answers (0 = yes; 1 = no); and (7) having taken hard drugs in the last year, including injected drugs, with 2 possible answers (1 = yes; 2 = no).

For each of these seven items, a new variable was created in order to compare those participants who had ever smoked (versus a nonsmoker), those who currently smoke tobacco (versus no), those considered a chain smoker (10+ cigarettes/day) (versus less than 10+ or none), those who reported moderate or high weekly average alcohol consumption (versus low/none), those who binge drink weekly or more often (versus less than weekly or none), those who had smoked marijuana in the past year (versus no), and those who had taken drugs in the last year (versus no).

### 2.2.6. Psychological Wellbeing and Illness

Psychological wellbeing was measured through 4 items: (1) The frequency of feeling down, depressed and hopeless in the last 2 weeks. This item had 4 possible answers (1 = not at all, 2 = several days, 3 = more than half of the days, and 4 = nearly every day) and was dichotomized (1 = more than half of the days or nearly every day; 0 = never or several days). (2) Screened for current depression. This item had 2 possible answers (1 = no; 1 = yes). (3) Currently taking medication for depression. This item had 2 possible answers (1 = yes; 1 = no). (4) The frequency of feeling apathetic in last 2 weeks. Like item 1, this had 4 possible answers (1 = not at all, 2 = several days, 3 = more than half of the days, and 4 = nearly every day) and was dichotomized (1 = more than half of the days or nearly every day; 0 = never or several days).

Illness was measured through 2 items: (1) Having a serious physical health infirmity. This item had 2 possible answers (1 = yes; 1 = no). (2) Having a longstanding limiting illness

or disability. This item had 3 possible answers (1 = none, 2 = not limiting, and 3 = limiting,) and was dichotomized into two categories (0 = limiting; 1 = not limiting or none).

### 2.2.7. Sociodemographics Factors

Information about the sex, age, ethnicity/race, civil marital status, religious affiliation, church attendance, residence area, and academic qualification of the participants was analyzed.

### 2.3. Procedure

The survey was conducted through face-to-face interviews and computer-assisted self-interviews (CASI). The interviews were conducted in the participants' homes or a private space in a mobile research vehicle. The interviewers used a random location sampling approach to recruit participants which involved selecting a random location in the selected sampling point and then approaching households in that area. Participants were also selected through a quota sampling approach to ensure a diverse sample in terms of age, sex/gender, ethnicity, and region of the UK. The survey questions covered a wide range of topics related to sexual attitudes, behaviors, and health, and the participants were assured of the confidentiality of their responses. A full description of the Natsal-3 procedures has been published [34].

### 2.4. Data Analyses

Weighted bivariate analyses (frequencies and percentages) were used to describe the main characteristics of the sample (Tables 1–3) and to show the distribution number of sexual partners in the last 5 years of concordant heterosexuals and closeted discordant heterosexuals by sex (Figure 1). Regression models assessed the association of each profile of participants with a dependent variable, comparing IBD-discordant heterosexuals—closeted, experimenters, and desisters—and nonheterosexuals with concordant heterosexuals (the reference group). All models were adjusted for sociodemographic factors (sex, age, residence area, ethnicity, academic qualifications, civil marital status, and religious affiliation). All but two outcomes were dichotomous, with one polytomous, resulting in logistic regression estimates presented as odd ratios (ORs) or relative risk ratios (RRRs). The number of sex partners was analyzed using linear regression, resulting in reported beta coefficients.

**Table 1.** Demographic characteristics of the sample in Natsal-3 by sex and sexual orientation, counts, and weighted proportions, Britain, 2010 (N = 15,162 participants).

| Characteristics | Total Population N % (CI) | | Males (N = 6293) N % (CI) | | | Females (N = 8869) N % (CI) | | | Heterosexual (N = 14,617) N % (CI) | | | Non-Heterosexual (N = 492) N % (CI) | | |
|---|---|---|---|---|---|---|---|---|---|---|---|---|---|---|
| Age (mean) | 15,162 | 43.10 (42.8–43.4) | 6293 | 42.88 | (42.5–43.3) | 8869 | 43.31 | (42.9–43.7) | 14,617 | 43.24 | (43.0–43.5) | 492 | 37.76 | (37.0–38.5) |
| Ethnicity/race | | | | | | | | | | | | | | |
| White | 13,351 | 88.73 (88.0–89.5) | 5551 | 88.26 | (87.1–89.3) | 7800 | 89.18 | (88.3–90.0) | 12,895 | 88.68 | (87.9–89.4) | 447 | 90.91 | (85.8–94.3) |
| Other | 1764 | 11.27 (10.6–12.0) | 717 | 11.73 | (10.7–12.9) | 1047 | 10.81 | (10.0–11.7) | 1710 | 11.31 | (10.6–12.1) | 44 | 9.08 | (5.7–14.2) |
| Academic qualifications | | | | | | | | | | | | | | |
| No qualifications | 2715 | 20.47 (19.6–21.3) | 1153 | 20.38 | (19.1–21.7) | 1562 | 20.57 | (19.5–21.6) | 2645 | 20.64 | (19.8–21.5) | 66 | 14.60 | (10.9–19.2) |
| Qualifications typically gained at age 16 years [†] | 4772 | 33.58 (32.7–34.5) | 1921 | 32.54 | (31.2–33.9) | 2851 | 34.59 | (33.4–35.8) | 4626 | 33.70 | (32.7–34.7) | 143 | 29.30 | (24.6–34.4) |
| Studying for or have attained further qualifications | 6808 | 45.95 (44.9–47.0) | 2851 | 47.07 | (45.5–48.7) | 3957 | 44.84 | (43.6–46.1) | 6540 | 45.65 | (44.6–46.7) | 261 | 56.11 | (50.2–61.8) |
| Marital status | | | | | | | | | | | | | | |
| Married | 5346 | 51.69 (50.7–52.7) | 2165 | 52.33 | (50.9–53.7) | 3181 | 51.06 | (49.7–52.4) | 5256 | 52.48 | (51.5–53.5) | 77 | 23.81 | (18.9–29.6) |
| Unmarried | 9520 | 48.31 (47.3–49.3) | 3987 | 47.67 | (46.2–49.1) | 5533 | 48.94 | (47.6–50.3) | 9103 | 47.51 | (46.5–48.5) | 405 | 76.18 | (70.4–81.1) |
| Religion | | | | | | | | | | | | | | |
| No religion | 7730 | 48.02 (47.0–49.0) | 3525 | 52.68 | (51.2–54.2) | 4205 | 43.43 | (42.2–44.7) | 7391 | 47.47 | (46.5–48.5) | 330 | 66.83 | (61.4–71.9) |
| Church of England | 2161 | 17.31 (16.6–18.1) | 762 | 14.78 | (13.8–15.9) | 1399 | 19.80 | (18.8–20.9) | 2124 | 17.58 | (16.8–18.4) | 37 | 8.41 | (6.1–11.6) |
| Other religions | 5223 | 34.67 (33.7–35.7) | 1980 | 32.52 | (31.1–34.0) | 3243 | 36.77 | (35.5–38.0) | 5087 | 34.96 | (34.0–36.0) | 125 | 24.76 | (20.1–30.1) |

Note: N, number of unweighted cases; CI, confidence interval. Denominators vary across variables because of item non-response. Estimates were weighted. Percentages and means were calculated by columns. [†] English General Certificate of Secondary Education or equivalent.

**Table 2.** Heterosexual population prevalence and size estimates of sexual identity/behavior subpopulations by sex, Britain, 2010 (N = 12,472).

| | Males N % (CI) | | | Females N % (CI) | | | *p* [a] |
|---|---|---|---|---|---|---|---|
| Concordant heterosexual | 4907 | 97.05 | (96.4–97.6) | 6631 | 95.75 | (95.2–96.3) | <0.001 |
| Discordant heterosexual | 144 | 2.95 | (2.4–3.6) | 351 | 4.25 | (3.8–4.8) | <0.001 |
| Closeted | 25 | 0.47 | (0.3–0.8) | 76 | 0.82 | (0.6–1.1) | 0.034 |
| Experimenter | 60 | 1.22 | (0.9–1.6) | 171 | 1.18 | (0.9–1.5) | 0.858 |
| Desister | 59 | 1.26 | (0.9–1.7) | 104 | 2.24 | (1.9–2.7) | <0.001 |
| **Total heterosexual** | 5051 | 100 | | 6982 | 100 | | |
| Heterosexuals | 5051 | 97.30 | (96.73–97.77) | 6982 | 97.16 | (96.72–97.55) | 0.694 |
| Non-heterosexuals | 172 | 2.70 | (2.23–3.27) | 241 | 2.84 | (2.45–3.27) | 0.694 |
| **Total males/females** | 5223 | 100 | | 7223 | 100 | | |

Note: [a] *p*-value of survey-adjusted Wald test. Population-weighted percentages and means are calculated by columns. Participants who did not report any sexual partners in the last 5 years (N = 2690) were excluded.

Based on preliminary modeling, outcome classifications were adjusted to form more interpretable or parsimonious categories in line with significant and substantive differences. All or most of the significant variation was captured by the following: for religious affiliation, the distinction between none and any affiliation; for education, the difference between no academic qualifications, the attainment of a General Certificate of Secondary Education (GCSE), and the pursuit or attainment of higher academic qualifications; for ethnicity, the distinction between white and all other ethnicities; and for age, the distinction between persons under the age of 45 and those 45 and older. This categorization of age also facilitates comparison with the Natsal-2 survey from 2000, which was limited to Britons under age 45.

Finally, combining answers from two items asking, "opinion of sexual relations between two adult men/women: right/wrong", a tolerance analysis of same-sex sexual relations was conducted, comparing concordant heterosexuals and IBD heterosexuals with nonheterosexuals by examining the possible influence of internalized homophobia on these opinions.

Statistical analyses for the present study were performed using Stata, versions 15.0 and 18.0, for Windows, adjusting for sample stratification, and clustering and weighting using information supplied by Natsal-3 so as to represent as closely as possible the British population of men and women aged 16–74 years. A significance level of 0.05 was established, and a significance level of 0.1 was informed for adjusted analyses.

**Table 3.** Sociodemographic characteristics of all analysis groups.

| | CH | Closeted (C) | Experimenter (E) | Desister (D) | NH | C vs. CH | E vs. CH | D vs. CH | NH vs. CH |
|---|---|---|---|---|---|---|---|---|---|
| | % (CI) | % (CI) | % (CI) | % (CI) | % (CI) | *p* [a] | *p* [a] | *p* [a] | *p* [a] |
| **Sex** | | | | | | | | | |
| Male | 51.0 (49.9–52.1) | 37.09 (25.3–50.6) | 36.52 (29.1–44.6) | 51.47 (42.4–60.4) | 49.40 (43.3–55.5) | <0.05 | <0.001 | 0.916 | 0.25 |
| Female | 49.02 (47.9–50.1) | 62.91 (49.4–74.7) | 63.48 (55.4–70.9) | 48.53 (39.6–57.6) | 50.60 (44.5–56.7) | | | | |
| **Age** | | | | | | | | | |
| 16–44 | 56.74 (55.6–57.9) | 56.42 (43.8–68.3) | 60.06 (51.4–68.2) | 58.42 (49.0–67.3) | 70.67 (64.7–76.0) | 0.960 | 0.445 | 0.724 | <0.001 |
| 45–74 | 43.26 (42.1–44.4) | 43.58 (31.7–56.2) | 39.94 (31.8–48.6) | 41.58 (32.7–51.0) | 29.33 (24.0–35.3) | | | | |
| **Ethnicity/race** | | | | | | | | | |
| White | 88.83 (88.0–89.6) | 91.47 (78.2–97.0) | 94.28 (89.0–97.1) | 90.97 (83.0–95.4) | 90.70 (85.0–94.4) | 0.547 | <0.01 | 0.475 | 0.428 |
| Other | 11.17 (10.4–12.0) | 8.53 (3.03–21.8) | 5.72 (2.9–11.0) | 9.03 (4.6–17.0) | 9.30 (5.6–15.0) | | | | |
| **Academic qualifications** | | | | | | | | | |
| No qualifications | 18.03 (17.1–19.0) | 6.28 (3.03–12.6) | 7.30 (4.3–12.1) | 11.07 (6.7–17.8) | 10.37 (7.3–14.5) | <0.001 | <0.001 | <0.05 | <0.001 |
| Qualifications typically gained at age 16 years [†] | 35.46 (34.4–36.5) | 34.31 (23.7–46.8) | 30.46 (23.9–37.9) | 32.2 (24.1–41.6) | 31.13 (26.1–36.7) | 0.848 | 0.168 | 0.474 | 0.124 |
| Studying for or have attained further qualifications | 46.51 (45.3–47.7) | 59.41 (47.0–70.7) | 62.24 (54.5–69.4) | 56.70 (47.5–65.5) | 58.49 (52.3–64.4) | <0.05 | <0.001 | <0.05 | <0.001 |
| **Marital status** | | | | | | | | | |
| Married | 54.86 (53.8–56.0) | 30.27 (19.1–44.3) | 49.06 (41.3–56.9) | 41.25 (32.4–50.7) | 25.3 (20.0–31.5) | <0.001 | 0.151 | <0.01 | <0.001 |
| Unmarried | 45.14 (44.0–46.2) | 69.73 (55.7–80.9) | 50.94 (43.1–58.7) | 58.75 (49.3–65.6) | 74.68 (68.5–80.0) | | | | |
| **Affiliation** | | | | | | | | | |
| No religion | 49.04 (47.9–50.1) | 66.18 (53.1–77.2) | 59.60 (51.5–67.2) | 57.44 (48.4–66.0) | 68.42 (62.7–73.7) | <0.01 | <0.01 | 0.065 | <0.001 |
| Church of England | 16.5 (15.7–17.4) | 12.9 (6.0–25.4) | 9.95 (5.6–17.0) | 12.45 (7.5–20.0) | 8.06 (5.7–11.3) | 0.446 | 0.021 | 0.201 | <0.001 |
| Other religion | 34.44 (33.4–35.5) | 20.96 (12.5–33.0) | 30.45 (23.9–38.0) | 30.11 (22.7–38.8) | 23.52 (18.6–29.3) | <0.05 | 0.275 | 0.297 | <0.001 |
| **Religious attendance** | | | | | | | | | |
| At least once a week | 16.50 (15.3–17.8) | 18.51 (6.0–44.7) | 14.87 (7.8–26.5) | 8.10 (3.6–17.0) | 17–47 (11.9–25.0) | 0.837 | 0.731 | <0.05 | 0.099 |
| Less than once a week | 83.50 (82.2–84.7) | 81.49 (55.3–94-0) | 85.13 (73.5–92.2) | 91.90 (83.0–96.4) | 82.53 (75.0–88.1) | | | | |
| **Residential area** | | | | | | | | | |
| Rural | 23.64 (22.1–25.2) | 27.06 (17.0–40.3) | 22.44 (15.2–31.9) | 23.54 (16.4–32.6) | 11.75 (8.4–16.3) | 0.567 | 0.778 | 0.981 | <0.001 |
| Urban | 76.4 (74.8–77.9) | 72.94 (59.7–83.0) | 77.56 (68.1–84.8) | 76.46 (67.4–83.6) | 88.25 (83.7–91.6) | | | | |
| **Sex of most recent live-in partner** | | | | | | | | | |
| Opposite sex | 99.91 (99.7–100.0) | 90.61 (74.8–96.9) | 99.36 (95.5–100.0) | 100 | 48.54 (36.2–53.8) | 0.069 | 0.394 | 0.075 | <0.001 |
| Same sex | 0.09 (0.0–0.3) | 9.39 (3.1–25.2) | 0.64 (0.1–4.5) | 0 | 55.15 (46.2–63.8) | | | | |
| **Number of same-sex partners in the past year** | | | | | | | | | |
| 0 | 100 | 0 | 100 | 100 | 35.45 (29.8–41.5) | - | - | - | <0.001 |
| 1 | 0 | 70.46 (58.1–80.4) | 0 | 0 | 41.22 (35.4–47.2) | <0.001 | - | - | <0.001 |
| 2 or more | 0 | 29.54 (19.6–41.9) | 0 | 0 | 23.33 (18.7–28.7) | <0.001 | - | - | <0.001 |

Excludes participants who did not report any sexual partners in the last 5 years (N = 2690). Abbreviations: CI, confidence interval; ref, reference; CH, Concordant Heterosexual; C, Closeted, E, Experimenter; D, Desister; NH, Nonheterosexual. [†] English General Certificate of Secondary Education or equivalent. [a] *p*-value of survey-adjusted Wald test. Population-weighted percentages are calculated by columns.

### 3. Results

*3.1. Descriptive for the Sample*

Characteristics of the population-representative sample are detailed in Table 1 according to sex and sexual orientation. The majority of the participants were female (51%) and heterosexual (97%). They also mainly identified as white (89%) and married (52%). Almost half of them were studying or had completed higher education (46%) and did not declare any religious affiliation (48%). These proportions were very similar for both males and females. Compared to heterosexual participants, nonheterosexual participants were more likely to be younger, with higher educational attainment, unmarried, and religiously unaffiliated compared to heterosexual ones.

The prevalence of participants belonging to the five groups of the analysis, stratified by sex, are detailed in Table 2. Of the 12,472 participants who reported having had a sexual partner in the last 5 years, significant differences ($p < 0.001$) by sex were reported by concordant heterosexuals (97% males vs. 96% females) and discordant heterosexuals (3% males vs. 4% females). Of the three groups of discordant heterosexuals, only desister discordants significantly differed with respect to sex (1% vs. 2%, $p < 0.001$). No differences by sex were found with respect to sexual orientation (heterosexuals vs. nonheterosexuals, $p = 0.694$).

*3.2. Sociodemographic Characteristics of All Analysis Groups*

Table 3 presents weighted estimates of sociodemographic characteristics among the five analyzed groups. Among concordant heterosexuals, 51% were male, with a mean age of 42 years, 89% were of white ethnicity, 47% had obtained higher academic qualifications, and most of them were married (55%), religiously affiliated (51%), and lived in an urban area (76%). Nearly 17% of concordant heterosexuals reported religious attendance. Almost all (99.9%) reported an opposite-sex partner as the most recent live-in partner, and none reported same-sex partners in the past year.

Among the discordant heterosexual groups, closeted individuals and experimenters were mostly females (61% and 63%, respectively), contrary to desisters (48%). The average ages ranged from 40 to 42 years. A great majority of participants of each group were white (92–95%), had obtained higher academic qualifications (59–62%), and lived in an urban area (72–78%). Conversely to concordant heterosexuals, they were mostly unmarried (51–69%) and did not report any religious affiliation (58%-69%). Attendance of religious services was low in all groups (8–19%). Among closeted discordants, 9% reported having a same-sex partner as the most recent live-in partner, and all of them reported having had a same-sex partner in the last year. Only 1% of experimenters reported a same-sex partner as the most recent live-in partner, whereas none reported same-sex partners in the last year. Among desisters, none reported having had a same-sex partner, either recently cohabiting or sexual, in the past year.

Nonheterosexuals constituted the youngest group (M = 36 years). Of them, 51% were female, 91% were white, 89% lived in an urban area, and 58% had obtained higher academic qualifications. They were more likely to be unmarried (75%). The proportion of nonheterosexuals with a religious affiliation was low, at levels similar to closeted discordants (68% and 69%, respectively), and 89% reported low religious attendance. As for the sex of their recent partners, 55% reported having had a same-sex partner in their last cohabitation, and 65% reported having had one or more same-sex sexual partners in the last year.

Table 4 shows adjusted associations for demographic characteristics, comparing each IBD group and nonheterosexuals with concordant heterosexuals.

**Table 4.** Adjusted odds ratios (ORs) or relative risk ratio (RRRs) for demographic characteristics, comparing non-heterosexuals (NH) and discordant heterosexual subgroups with concordant heterosexuals (CH).

| | C vs. CH | E vs. CH | D vs. CH | NH vs. CH |
|---|---|---|---|---|
| | OR or RRR (95% CI) [a] | OR or RRR (95% CI) [a] | OR or RRR (95% CI) [a] | OR or RRR (95% CI) [a] |
| Female (ref = male) | 1.79 * (1.0–3.2) | 1.84 ** (1.3–2.6) | 1.00 (0.7–1.5) | 1.13 (0.9–1.5) |
| Age (16–44; 45–74; ref = younger) | 2.00 * (1.1–3.5) | 1.19 (0.8–1.8) | 1.33 (0.9–2.0) | 1.02 (0.8–1.4) |
| White ethnicity (ref = other) | 1.56 (0.4–6.4) | 2.09 [1] (1.0–4.5) | 1.17 (0.5–2.5) | 1.16 (0.7–1.9) |
| Academic qualifications (ref = no qualifications) | | | | |
|     Qualifications typically gained at age 16 years [†] | 3.09 * (1.3–7.3) | 2.10 * (1.2–3.8) | 1.53 (0.8–2.8) | 1.44 [1] (0.9–2.2) |
|     Studying for or have attained further qualifications | 4.51 ** (1.9–10.7) | 3.49 *** (1.9–6.3) | 2.10 * (1.2–3.8) | 2.04 ** (1.3–3.1) |
| Married (ref = unmarried) | 0.35 ** (0.2–0.6) | 0.87 (0.6–1.2) | 0.56 ** (0.4–0.8) | 0.34 *** (0.2–0.5) |
| Has a religious affiliation (ref = none) | 0.46 * (0.2–0.9) | 0.64 ** (0.5–0.9) | 0.77 (0.5–1.1) | 0.55 *** (0.4–0.7) |
| Urban residence (ref = rural) | 0.80 (0.4–1.5) | 1.19 (0.8–1.8) | 0.98 (0.6–1.6) | 2.23 *** (1.5–3.3) |

Odds ratios or relative risk ratio (95% confidence intervals) for each characteristic are adjusted for all other characteristics shown. Asterisks report *p*-values of *t*-tests for which the OR is not equal to 1: [1] $p < 0.1$; * $p < 0.05$; ** $p < 0.01$; *** $p < 0.001$ Abbreviations: OR, odds ratio; RRR, relative risk ratio; CI, confidence interval; ref, reference; CH, Concordant Heterosexual; C, Closeted, E, Experimenter; D, Desister; NH, Nonheterosexual. [†] English General Certificate of Secondary Education or equivalent. [a] Excludes 2940 respondents (19%) that did not report their outcome.

Group A: Closeted Discordant Heterosexuals:

Compared to concordant heterosexuals, closeted discordant heterosexuals were nearly twice as likely to be female and older, up to three times more likely to have postsecondary education, and more than four times more likely to have further academic qualifications. They also were two-thirds less likely to be married and nearly half less likely to report a religious affiliation.

Group B: Experimenter discordant heterosexuals:

Experimenter discordants, compared to concordant heterosexuals, were more likely to be female, twice as likely to have had post-secondary education, and more than three times as likely to have further academic qualifications. Also, they were one-third less likely to be married.

Group C: Desister discordant heterosexuals:

Desister discordant heterosexuals were twice as likely to have further academic qualifications than concordant heterosexuals, whereas they were nearly half as likely to be married.

Group D: Nonheterosexuals:

Nonheterosexuals were twice as likely to have further academic qualifications and an urban residence. They also were two-thirds less likely to be married and nearly half as likely to report a religious affiliation.

In summary, the closeted discordants and experimenters were more likely to be women. The experimenters were also younger. The nonheterosexual group was more likely to live in an urban area. All groups reported a higher likelihood of having more educational attainment, compared to the heterosexual closeted. In addition, all groups reported lower

odds of being married except the experimenters. Finally, all were less likely to belong to a religion.

### 3.3. Association between Attitudes and Behaviors and Each of the Five Groups of Interest

Table 5 overviews adjusted models for selected risk behaviors and attitudes, comparing each group of discordant heterosexuals and nonheterosexuals with concordant heterosexuals.

**Table 5.** Adjusted odds ratios (ORs) for selected behaviors and attitudes, comparing identity/behavior discordant heterosexuals and non-heterosexuals with concordant heterosexuals. Britian, 2010 (N = 12,472).

| | C vs. CH | E vs. CH | D vs. CH | NH vs. CH |
|---|---|---|---|---|
| | OR (95% CI) [a] | OR (95% CI) [a] | OR (95% CI) [a] | OR (95% CI) [a] |
| **Opinions about Sexuality** | | | | |
| High pro-LGBT attitudes (ref = low) [c] | 1.75 [1] (0.9–3.3) | 1.29 (0.9–1.8) | 1.30 (0.9–1.9) | 5.43 *** (4.0–7.4) |
| Same-sex sexual relations are always wrong (refcat = non-heterosexuals) | | | | |
| Same sex sexual relations are never wrong (refcat = non-heterosexuals) | | | | 1.0 (reference) |
| Adultery is rarely or never wrong (ref = mostly or always wrong) | 3.77 * (1.3–10.8) | 2.40 [1] (1.0–6.0) | 3.81 ** (1.5–9.7) | 2.45 ** (1.3–4.8) |
| One-night stands are rarely or never wrong (ref = always wrong) | 1.22 (0.7–2.1) | 1.58 * (1.1–2.4) | 1.48 [1] (1.0–2.3) | 1.35 * (1.0–1.7) |
| Sex without love is OK; agree strongly (ref = else) | 1.62 (0.9–3.0) | 2.08 *** (1.4–3.1) | 2.54 *** (1.6–4.0) | 1.54 (1.1–2.1) |
| SEXUAL RISK BEHAVIORS | | | | |
| Linear trend in the number of sex partners | 15.00 *** (8.2–27.3) | 2.03 *** (1.5–2.8) | 4.08 *** (2.5–6.6) | 2.95 *** (2.3–3.9) |
| High STI risk (ref = low or none) | 4.91 *** (2.4–10.0) | 0.62 (0.3–1.5) | 3.13 ** (1.5–6.6) | 3.58 *** (2.2–5.7) |
| Ever paid for sex (ref = never) | 3.13 * (1.3–7.9) | 2.08 [1] (0.9–5.0) | 7.75 *** (4.4–13.8) | 1.48 (0.8–2.7) |
| SUBSTANCE USE | | | | |
| Ever smoked tobacco (ref = never) | 2.14 ** (1.3–3.6) | 3.02 *** (2.1–4.4) | 2.20 *** (1.5–3.3) | 1.59 *** (1.2–2.1) |
| Currently smokes tobacco (ref = no) | 2.29 ** (1.3–4.0) | 1.83 ** (1.3–2.6) | 1.43 [1] (1.0–2.2) | 1.46 ** (1.1–1.9) |
| Chain smoker (10+ cigarettes/day) (ref = less than 10 or none) | 1.59 (0.8–3.2) | 1.71 * (1.0–2.8) | 1.44 (0.8–2.5) | 1.15 (0.8–1.7) |
| Moderate/high weekly alcohol use (ref = none/low) | 1.83 [1] (1.0–3.4) | 1.63 * (1.0–2.6) | 1.95 ** (1.2–3.1) | 1.60 ** (1.2–2.2) |
| Binge drinks weekly or more often (ref = less than weekly or none) | 1.99 * (1.1–3.7) | 1.32 (0.9–2.0) | 1.30 (0.8–2.1) | 0.96 (0.7–1.3) |
| Smoked marijuana in the past year (ref = no) | 2.74 ** (1.5–5.0) | 2.04 ** (1.3–3.3) | 2.19 ** (1.4–3.5) | 1.49 * (1.1–2.0) |
| Used hard drugs in the past year (ref = no) | 4.14 *** (2.1–8.3) | 3.48 *** (2.0–6.1) | 2.91 *** (1.7–4.9) | 3.86 *** (2.7–5.5) |
| **Psychological Wellbeing and Illness** | | | | |
| **Feeling depressed in the last 2 weeks** (ref = not at all) | 1.08 (0.6–2.1) | 0.97 (0.6–1.7) | 1.28 (0.8–2.2) | 2.20 ** (1.6–3.1) |
| **Screened positive for current depression** (ref = screened negative) | 0.86 (0.5–1.6) | 1.04 (0.6–1.7) | 1.11 (0.7–1.9) | 1.75 ** (1.3–2.5) |
| **Currently taking medication for depression** (ref = no) | 1.41 (0.7–2.8) | 2.49 *** (1.6–3.9) | 1.57 (0.8–3.0) | 2.52 *** (1.8–3.6) |
| **Feeling apathy in the last 2 weeks** (ref = not at all) | 0.97 (0.5–1.9) | 1.02 (0.6–1.8) | 1.01 (0.6–1.8) | 1.62 * (1.1–2.3) |
| **Has serious physical health infirmity** (ref = no) | 0.65 (0.3–1.5) | 1.13 (0.7–1.8) | 1.58 [1] (1.0–2.6) | 1.51 * (1.0–2.3) |
| **Has longstanding limiting illness or disability** (ref = non-limiting or none) | 0.89 (0.5–1.8) | 1.52 * (1.0–2.3) | 1.53 [1] (0.9–2.5) | 2.52 *** (1.9–3.4) |

Excludes participants who did not report any sexual partners in the last 5 years (N = 2690). Asterisks report the *p*-value of *t*-tests for which the OR is not equal to 1: [1] *p* < 0.1; * *p* < 0.05; ** *p* < 0.01; *** *p* < 0.001. Abbreviations: OR, odds ratio; CI, confidence interval; ref, reference; CH, Concordant Heterosexual; C, Closeted; E, Experimenter; D, Desister; NH, Nonheterosexual. [a] Adjusted for sociodemographic factors: age, sex, residence area, ethnic identity, academic qualifications, civil marital status, and religious affiliation. [c] A single measure of pro-LGBT attitudes was created, obtaining the mean score of the four attitudes. Internal consistency was very good (Cronbach's alpha = 0.90). This variable was dichotomized by the median to create high and low groups.

Opinions about sexuality and LGBT rights:

Compared to concordant heterosexuals, nonheterosexuals were 5.4 times more likely to support pro-LGBT attitudes in favor of same-sex relationships and adoption rights. Closeted heterosexuals, at 1.8 times more likely to support pro-LGBT attitudes, were significantly higher than concordant heterosexuals at the 0.10 critical level but not at 0.05. Experimenters and desisters were no more likely than concordant heterosexuals to express pro-LGBT attitudes. Closeted and desisters were more likely to report that "adultery is rarely or never wrong", while experimenters and nonheterosexuals were more likely to report that "one-night stands is rarely or never wrong". Lastly, experimenters and desisters strongly agreed that sex without love is okay.

Sexual risk behaviors:

All groups reported a higher number of sexual partners compared to the heterosexual matched group, but undoubtedly, the closeted group reported a much higher number of sexual partners than the rest. Figure 1 displays the number of sexual partners in the last 5 years of closeted heterosexuals and concordant heterosexuals by sex. As can be seen, the distribution of the number of partners is reversed for each group. Among concordants, the higher the number of partners, the lower the proportion of heterosexual concordants. The opposite occurs among closeted discordants. Both trends are very similar among men and women.

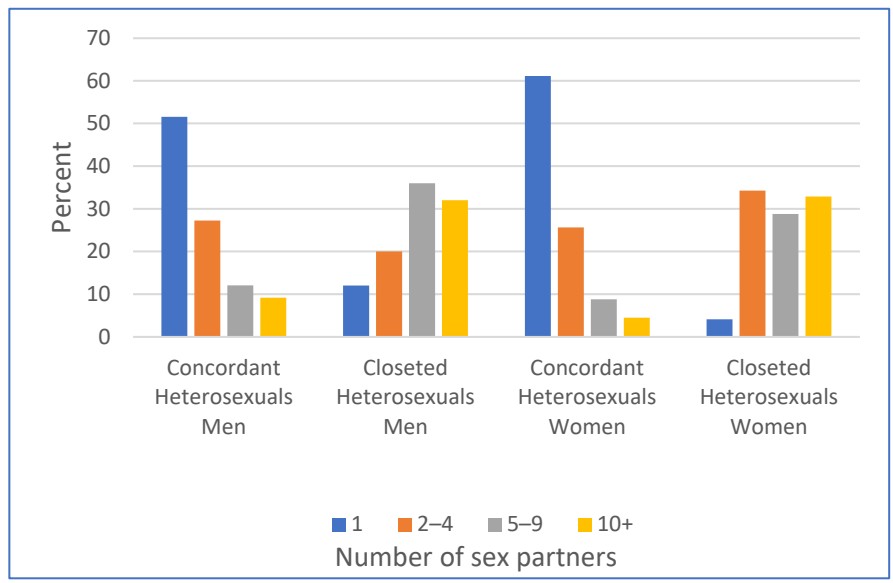

**Figure 1.** Number of sexual partners in the last 5 years of concordant heterosexuals and closeted discordant heterosexuals by sex.

On the other hand, the reported risk of STIs was significantly higher among the closeted, the desisters, and the nonheterosexuals. Only the closeted and desisters were more likely to have ever paid for sex.

Substance use:

Overall, all groups reported a higher likelihood of substance use (tobacco, alcohol, binge drinking, marijuana, and hard drugs) compared to concordant heterosexuals. The highest ORs were found for using hard drugs in the past year for all groups. Only the experimenter group reported a higher risk of being a chain smoker and the closeted group of weekly binge drinking.

Psychological wellbeing and illness:

Among heterosexual discordants, only members of the experimenter group were more likely than concordant heterosexuals to be currently taking medication for depression

and to have a longstanding limiting illness or disability. Not surprisingly, the studied indicators of reduced psychological wellbeing (i.e., depression, apathy) and physical health (i.e., physical infirmity, limiting illness or disability) were much more common among nonheterosexual persons than among heterosexuals.

Tolerance Analysis:

As noted above, despite their exposure to same-sex behavior, none of the IBD groups expressed more support for same-sex relationships and adoptions than did concordant heterosexuals, although the closeted group came close. To address the possibility of IBD homophobia, we examined the extreme responses to the questions asking for the respondents' views as to whether same-sex sexual relations between adults were right or wrong. The possible responses were always wrong, mostly wrong, sometimes wrong, rarely wrong, and not wrong at all. "Always wrong" expressed the most intolerant response; "not wrong at all" the most tolerant response. Table 6 reports the odds for the most tolerant and most intolerant responses for all four heterosexual groups—concordant heterosexuals and the three IBD groups—compared to nonheterosexuals.

**Table 6.** Tolerance analysis: adjusted odds ratios (ORs) for intolerance or tolerance of same-sex sexual relations, comparing identity/behavior-discordant heterosexuals and concordant heterosexuals with non-heterosexuals. Britian, 2010 (N = 12,472).

| | CH vs. NH | C vs. NH | E vs. NH | D vs. NH |
|---|---|---|---|---|
| | OR (95% CI) [a] | OR (95% CI) [a] | OR (95% CI) [a] | OR (95% CI) [a] |
| **Same-sex sexual relations always wrong** [b] | 6.26 *** (2.45–16.0) | 0.93 (0.20–4.4) | 1.59 (0.47–5.41) | 2.32 (0.70–7.64) |
| **Same-sex sexual relations never wrong** [b] | 0.19 *** (0.12–0.28) | 0.33 ** (0.15–0.71) | 0.23 *** (0.13–0.39) | 0.25 *** (0.14–0.45) |

Excludes participants who did not report any sexual partners in the last 5 years (N = 2690). Asterisks report *p*-value of *t*-tests for which the OR is not equal to 1: [1] $p < 0.1$; * $p < 0.05$; ** $p < 0.01$; *** $p < 0.001$. Abbreviations: OR, odds ratio; CI, confidence interval; ref, reference; CH, Concordant Heterosexual; C, Closeted, E, Experimenter; D, Desister; NH, Nonheterosexual. [a] Adjusted for sociodemographic factors: age, sex, residence area, ethnic identity, academic qualifications, civil marital status, and religious affiliation. [b] Combined answers from two items asking, "opinion of sexual relations between two adult men/women: right/wrong", with response options of always wrong, mostly wrong, sometimes wrong, rarely wrong, and not wrong at all. "Always wrong" reports those responding "always wrong" to either item (1) versus all other responses (0). Never wrong reports those responding "not wrong at all" (1) to either item versus all other responses (0).

As Table 6 reports, all heterosexual groups, including all three IBD groups, were significantly less likely than nonheterosexuals to report high tolerance of same-sex sexual relations, with ORs ranging from 0.19 to 0.33. On the other hand, while concordant heterosexuals were 6.3 times more likely to express high intolerance of same-sex sexual relations, none of the discordant heterosexual groups expressed higher intolerance than did nonheterosexuals. The lack of support for pro-LGBT attitudes among IBD heterosexuals is not linked to high intolerance of same-sex sexual behavior consistent with possible homophobia, as is the case among concordant heterosexuals.

## 4. Discussion

To our knowledge, this is the first study to explore the lifetime fluidity of discordance among currently identified heterosexual persons. The results reveal three distinct IBD patterns among such persons, who differ markedly from one another on indicators of health status, risk behaviors, and attitudes toward sexuality.

Our results confirm numerous previous findings that sexual fluidity was higher among women [1,18,38,39]. Many studies have identified this phenomenon as true among homosexuals. For example, Vrangalova and Savin-Williams [26] found that men tended to give more consistent responses to their sexual orientation than women. Ott et al. [39] found that women reported higher mobility scores regarding sexual orientation identity than

men. Findings from a recent longitudinal study of US adults demonstrated that women and nonheterosexuals reported a more fluid sexual identity than heterosexual men [39]. Our study adds to the few who have extended this finding also to heterosexuals, finding that 4.3% of heterosexual women, compared to 3.0% of heterosexual men, experienced IBD. These higher rates of IBD among heterosexual women than heterosexual men are also consistent with those found in prior studies [22–25].

The origin of these sex disparities, as pointed out by Diamond [17], remain uncertain and could be related to sexual minority status. While changes can manifest in either direction, there tends to be greater stability among those who self-identify as heterosexual compared to those who identify as homosexual, bisexual, or even mostly heterosexual [39–41]. Large longitudinal studies have also shown these changes are more often oriented toward heterosexuality than homosexuality [1]. In line with previous studios, our investigation demonstrates that incongruities and fluidity of the sexual orientation are more prevalent among women and emphasizes that these aspects are not restricted solely to sexual minority groups.

IBD discordance: closeted, experimenters, and desisters:

When considering the three heterosexual IBD groups collectively in comparison to the nonheterosexual group, several notable similarities and dissimilarities emerge. The heterosexual IBD groups tended to share with nonheterosexuals more secular and less moralistic characteristics than the general heterosexual population: more advanced education, lower participation in marriage and religion, and higher support for adultery and one-night stands. The discordant heterosexuals were also not more negatively proscriptive of same-sex sexual relations than were nonheterosexuals, but at the same time, they were not more positively affirming of them than were concordant heterosexuals. Like nonheterosexuals, all heterosexual IBD groups also reported more lifetime sex partners than did concordant heterosexuals and were also more prone to substance abuse, including present and past smoking and the use of alcohol, marijuana, and hard drugs. Nonheterosexuals, however, were more likely to reside in urban areas and to be more supportive of same-sex relationships and adoption rights and were more susceptible to feelings of depression and apathy than were the heterosexual IBD groups. Each IBD group also presents a unique set of characteristics relative to concordant heterosexuals compared to other IBD groups and to the nonheterosexual group.

Experimenters:

Experimenters, who had had only a single same-sex sexual partner more than a year ago and only heterosexual partner(s) in the past year, reported the least risky sexual behavior—the fewest sex partners, the lowest STI risk, and the least recourse to prostitution—among the three IBD groups. Like the other IBD groups, experimenters reported more accepting attitudes toward one-night stands and adultery but also toward loveless sex. Like the closeted group, the experimenter group was more female, and experimenters were the only IBD group that were significantly more likely to be of white race or ethnicity than concordant heterosexuals.

Experimenters had similar levels of psychological and physical well-being as the concordant heterosexual population on most measures but were more likely to have a limiting illness/disability or to be taking medication for depression. Experimenters also did not differ from concordant heterosexuals in terms of their pro-LGBT attitudes or other variables such as marital status, age, and area of residence. However, the experimenters were more similar to the nonheterosexual group, in contrast to concordant heterosexuals, with respect to their higher educational level, substance abuse, low religious affiliation, and support for one-night stands and adultery.

Closeted:

Closeted IBD heterosexuals, who reported ongoing same-sex sexual encounters often contiguous with heterosexual partnerships, stand out for both their elevated number of

sexual partners in the past five years and their increased risk for STIs. Closeted heterosexuals reported fifteen times more sex partners and an almost five times higher STI risk, on average, than did concordant heterosexuals. This is in stark contrast to patterns observed in the heterosexual population and substantially higher than among nonheterosexuals, experimenters or desisters.

Like experimenters, closeted IBD heterosexuals were more female than either desisters or nonheterosexuals. The preponderance of women among experimenters and closeted IBDs may suggest that as a form of sex exploration, women may more frequently engage in sexual activities with same-sex partners without leading them to question their heterosexual identity. Heterosexual identification despite continued same-sex activity may be due to the avoidance of stigma; however, in this case, one might have expected a higher proportion of males relative to females among the closeted, given that males tend to report greater social stigma [42] and higher levels of internalized homonegativity [43].

Closeted heterosexuals are similar to concordant heterosexuals in terms of ethnicity, urban residence, levels of psychological well-being and illness, and sexual attitudes except for adultery. Conversely, they are more similar to the nonheterosexual group in reporting higher academic qualifications, low religious affiliation, being unmarried, and higher substance use. Of the IBD groups, the closeted had the highest support for same-sex relationships and the adoption of children by gays and lesbians, significantly different from concordant heterosexuals at the 0.10 critical level but still less than half as high as nonheterosexuals.

Desisters:

Desisters, who reported significant past same-sex sexual behavior but none in the past year, were the most similar to concordant heterosexuals of the three heterosexual IBD groups. Desisters generally matched concordant heterosexuals demographically in terms of sex, age, ethnicity, religious affiliation, and area of residence. Their psychological profile was similar to concordant heterosexuals, and like them, they did not report high pro-LGBT attitudes. Unlike concordant heterosexuals and like nonheterosexuals, however, desisters reported higher acceptance of loveless sex, one-night stands, and adultery and higher substance use and risky sexual behaviors.

Unlike experimenters and the closeted group, desisters were not more likely to be female. In addition, they were the group most likely to have ever paid for sex, consistent with their greater male composition.

Nonheterosexual group:

Some of the most striking findings of this study were those related to the nonheterosexual group. In terms of prevalence, we found similar rates between males and females, ranging from 2 to 3%. These proportions are similar to those reported by the UK Census 2021, according to which, the proportion of the UK population aged 16 years and over identifying as lesbian, gay, or bisexual (LGB) in 2020 was 3.1% [36]. This survey collected information about sexual orientation using a question designed to capture sexual identity comparable to the one in this study.

Nonheterosexuals reported much more strongly favorable attitudes toward same-sex relationships and the adoption of children by gays and lesbians than did heterosexual IBDs. Closeted heterosexuals, who currently engaged in same-sex behavior, were more supportive of these pro-LGBT attitudes than experimenters or desisters who had ceased doing so, but none of the heterosexual IBD groups were even half as likely to support them as the nonheterosexuals were. The absence of negative judgment of same-sex behavior among heterosexual IBDs argues against a simplistic attribution of these differences to homophobia or to homophobia alone. Rather, these facts emphasize the close correlation between opinions and self-identification, suggesting that adopting a sexual identity may not simply be a way of viewing oneself but also a way of viewing the cultural and political world in which one lives. We already suggested above that sexual attraction and sexual

behavior might share underlying mechanisms that could be distinct from those shared by self-identification and opinions.

With some exceptions, nonheterosexual individuals also reported worse physical and psychological health than heterosexual IBDs. Compared to concordant heterosexuals, experimenters were as likely as nonheterosexuals to be on depression medication, desisters were as likely to have a physical infirmity, and both experimenters and desisters were at an elevated but lower risk of a limiting illness or disability. This finding extends abundant previous research which found homosexual individuals to be at a higher risk of depression, anxiety, substance abuse, and suicidal ideation compared to their heterosexual counterparts [2,24,44]. Our data do not allow us to know the reasons for which these persons reported poorer health, but they do highlight that the nonheterosexual population is more vulnerable to health problems than heterosexuals, discordant or concordant. Many studies corroborate that sexual minority individuals may experience a disproportionately higher prevalence of adverse childhood experiences (ACEs), increasing their exposure to multiple developmental risk factors that have systemic negative health effects across their lifespan [45]. Supporting this explanation, our results also showed higher substance use and sexual risk behaviors of this group compared to heterosexual concordants.

According to the minority stress theory, "concealment" refers to the practice of hiding or suppressing one's sexual minority identity in order to avoid the potential negative effects of stressors such as discrimination or stigma [46]. While concealment may provide some short-term relief from stressors, it can also lead to additional stress in the long term as it requires continuous effort to maintain concealment and may result in a lack of authenticity in one's interactions and relationships [47]. From this theory, we would expect to have found worse health indicators among heterosexual discordants, particularly among those categorized as closeted. On the contrary, we found that discordant heterosexuals did not present greater psychological and physical problems than concordant heterosexuals, subject to the few exceptions already noted, none of which included the closeted. It is possible that the identity reported in the study was different than that publicly expressed by the participants, or that IBD health decreases over time due to concealment. Future research could explore these possibilities further.

*Limitations and Concerns*

Some limitations and cautions for interpreting the present study should be noted. Small group sizes prevented us from exploring many theoretically important covariates or associations of heterosexual IBD, such as race and sex, and from examining IBD among nonheterosexuals. We did not examine sexual attraction, a component of sexual orientation which may affect IBD in ways that we did not observe. The Natsal-3 measures employed self-reported retrospective data which are subject to several sources of bias, in particular social desirability and recall bias. In particular, we cannot be sure that the sexual identity reported by participants on the Natsal-3 survey is the same as that disclosed in other life settings such as with family and friends. Finally, as with any cross-sectional observational study, causal relationships between the factors studied cannot be determined from the data alone.

## 5. Conclusions

This study is the first to examine heterosexual identity–behavior discordance in Great Britain. The findings reveal a more diverse relationship between nonheterosexual sexual behavior and identity than is commonly perceived, which can contribute, in practice, to a better assessment of the health and lifestyle differences among the study groups observed and, in theory, to a better understanding of sexual orientation itself.

Our findings exemplify a more comprehensive and open-ended expression of nonheterosexual sexual orientation that is not confined to populations defined as homosexual or bisexual. Indeed, the population share of discordant heterosexuals (4.3%) is about half again larger than that of gays, lesbians, and bisexuals combined (2.9%; see Table 2).

Whatever sexual orientation may be conceived to be, it must be understood in a broad extensional sense that incorporates the complex and unbounded interactions of sexual fluidity and discordance, as well as transience, questioning, and uncertainty.

Practically, the IBD groups manifested unique health characteristics and attitudes. Relative to concordant heterosexuals, discordance was associated with more permissive social attitudes regarding adultery, sex without love, and one-night stands as socially permissible behaviors. It was also associated with higher substance abuse and risky sexual behaviors, particularly a higher number of sexual partners, particularly among the closeted. The physical and mental health of IBD heterosexuals was more similar to concordant heterosexuals than to nonheterosexuals, who consistently reported poorer physical and psychological health status.

This information could be useful for developing targeted interventions, policies, and programs to address disparities and improve outcomes for discordant heterosexual and nonheterosexual populations. Clinicians should be aware that sexual identity does not always correspond to assumptions regarding sexual behavior or attraction. In particular, assessing sexual behavior history for those identifying as heterosexual would enable a better understanding of their specific needs, allowing for more effective interventions tailored to different groups.

For counseling and psychotherapy, this would involve training counselors and mental health professionals in competence to effectively support discordant individuals by offering counseling services that respect their unique cultural and religious backgrounds and address any conflicts related to sexual orientation. It would be advisable for support services to develop resources through which closeted individuals could access reliable information about sexual health, including STI screenings. Ensuring that mental health services are socially accessible and available is as important for IBD heterosexuals as it is for all nonheterosexual persons. Finally, research efforts to better understand the health needs of nonheterosexual persons should include, as far as possible, IBD heterosexuals as well as self-identified gay, lesbian, or bisexual persons.

**Author Contributions:** Conceptualization, D.P.S. and S.J.; formal analysis, M.C. and D.P.S.; investigation, M.C. and D.P.S.; writing—original draft, M.C. and D.P.S.; writing—review and editing: S.J., M.C. and D.P.S. All authors have read and agreed to the published version of the manuscript.

**Funding:** This research received no external funding.

**Institutional Review Board Statement:** The present study's anonymous secondary analysis of these pre-existing public data was approved by the University of Navarra Institutional Review Board (reference: 2023.153 on 27 July 2023). The Natsal-3 study was conducted in accordance with the Declaration of Helsinki, and approved by the Institutional Review Board of The Oxfordshire Research Ethics Committee A (reference: 09/H0604/27).

**Informed Consent Statement:** Informed consent was obtained from all subjects involved in the Natsal-3 study.

**Data Availability Statement:** The data presented in this study are openly available in UK Data Service at 10.5255/UKDA-SN-7799–2, reference number 7799.

**Conflicts of Interest:** The authors declare no conflict of interest.

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
