# Peer review of "Sexual Identity–Behavior Discordant Heterosexuals in Britain: Findings from the National Survey of Sexual Attitudes and Lifestyle 2010–2012 (Natsal-3)"

_sexes, doi:10.3390/sexes4040039_

Round 1
Reviewer 1 Report
Comments and Suggestions for Authors
The article under review concerns the actual question of heterosexual attraction-behavior-identity con-/discordance. This study compares favorably with others in that it includes all dimensions of sexuality in the analysis. Indeed, we often ignore the fact that heterosexuals with a mismatch between the three dimensions of sexuality often fall into one wide category of bisexuals. For this reason, discordant minorities are often out of the research focus.
Objective of the study is to better describe the phenomenon of fluidity of heterosexual identity-behavior discordance, and to contribute new information regarding this population. This objective is successfully gained. It is also noteworthy that the authors classify three distinct profiles of discordant heterosexuals—closeted, experimenters and desisters. New findings obtained from national-level interviews based on a stratified multistage cluster sampling are very interesting and reliable, especially the facts that a) some forms of discordance between sexual identity and behaviour – concealing and experiment – are more salient in women than in men; b) desisters prevail in IBD heterosexual men; c) discordant straight individuals are mostly unmarried, in contrast with concordant ones.
This study adds to the few research on sexual fluidity and IBD among heterosexual women.
I consider very interesting the explanation, made by the authors, that adopting a sexual identity may not be simply a way of viewing oneself, but also a way of viewing the cultural and political world in which one lives.
As comments, I would like to mention the following.
In Discussion section, some peculiarities of experimenters group seem to be the artificial facts because it is not explained why such factor as “a single same-sex sexual partner more than a year ago” correlates with a limiting illness/disability or taking medication for depression (lines 616-617).
There are also some errors in the text:
Lines 112, 591 – misprint: IDB should be changed into IBD (identity-behavior discordance).
Lines 119, 126 – I would avoid duplication in combination of words: instead of “same-sex sex” I would write “same-sex encounters / practices / behaviors”.
Lines 171/172 – most likely, duplication of the definition “non-heterosexual sexual behavior” is not consonant: I would prefer “non-heterosexual behavior”.
In lines 262/263 – to understand the logic of the dichotomous division of responses on the Likert scale (1-4 = 0 and only 5 = 1), it is necessary to indicate in the text the scale values for each sign and explain why 4 was not combined with 5.
Lines 282/283 – I would recommend, in first, decrypt “pw” as “per week”, since not everyone may be familiar with such an abbreviation.
I found a discrepancy between the text in the lines 365/366 (only experimenter discordants significantly differed respect to sex (1% vs. 2%, p< .001) and data presented in the Table 2 (desister discordants significantly differed respect to sex (1% vs. 2%, p< .001)
Line 505 I would recommend choosing either "IBDs" or "discordants" to avoid doubling.
Line 581: “heterosexual” should be changed to “homosexual” (there is a mistake in the text).
Comments on the Quality of English LanguageThere are some stylistic flaws mentioned in the Comments and Suggestions
Author Response
Please find attached our response to your comments. Please note that in the manuscript you will find the changes suggested by reviewer 2 and by you (reviewer 1).

Reviewer 2 Report
Comments and Suggestions for Authors
Calatrava et al., present their findings on the analysis of the Natsal-3 on sexual identity and behavior in a sample of Britain heterosexual people. Analysing the attitudes towards ‘sexuality’, risk behaviors and health indicators of discordant heterosexual individuals into three categories, they showed that discordance was not evenly distributed across categories, and associated differently with the aforementioned variables across groups.
Overall, the study has its merits beyond the large sample to which the authors had access. Whereas the study is mainly descriptive, with some inferential statistics analyses through Odds Ratio’s, their manuscript presents a cohesive story, yet with several, though mostly minor, details that are either missing or need to be corrected.
Authors present a well-crafted introduction. It flows well and includes all relevant topics in a good argumentative line. It is worth noticing that authors do over use references 11 and 12 to state several points. Also, whereas the introduction is comprehensive, it also comes across a bit too long. Finally, it lacks a problematization, making the connection between the last paragraph and the “objective” abrupt.
In the method section, authors do an overall good describing the necessary information. There are few things they need to clarify in order to avoid confusion.
Similarly, there are several aspects to be corrected and improved in the results section. In particular, it is concerning that authors seem to believe they can just set an alpha level, and they choose which non-significant “marginal” results to report. It is a ‘pick-and-choose’ practice that should be avoided in order to minimize the risk of the family-wise error.
In the discussion and conclusion, authors do a good job to cover all results. However, the discussion lacks of sustenance in many parts. There are few external studies utilized to discuss their findings, many repeated, many more absent. The limitations stated are just not true limitations. Finally, in the conclusions, they talk about “developing policies, and programs”, yet nothing of that is discussed, remaining only on the surface the word can give, whatever these policies and programs they suggest could be.
Finally, there are a few inconsistencies across the manuscript in terms of right use of citations. Authors need to revise this in their second version.
Altogether, the manuscript requires a major review from the authors to re-submit a version suitable for publication. Therefore, I hereby suggest the manuscript be reconsidered after major revisions. I hope the authors may improve their manuscript with the suggestions below, and likely from other reviewers.
Sincerely,
Title
- I believe it would be necessary to include “Sexual” at the beginning of the title, as it is a way the phenomenon is described in the literature.
- It would serve well the authors and readers that the title includes “Natsal-3” as it is an abbreviation used for the studies steaming from that sample.
Abstract and Keywords
No comments
Introduction
Line 45 = what about bisexuals? Scores of 2 to 4 are generally accepted to comprise the bisexual range of the Kinsey scale.
see Rieger G. et al.., Sexual arousal: The correspondence of eyes and genitals. Biol. Psychol. 104, 56–64 (2015).
Line 57 = “does not accurately capture discordant sexual minorities” nor do people questioning their sexual orientation.
Regarding the intro 2nd paragraph, whereas it may apply better for paraphilias, I believe authors should at least mentioned that sexual orientation as described, for instance, by Seto 2016, encompasses by much more than attraction, behavior, and identity. Seto’s model is much more comprehensive model, and much like authors discuss the limitations of previous research as to how measure sexual attraction/orientation, I believe a more comprehensive model should be, at least, acknowledged. This is also true when speaking about sexual minorities, for these are not just gender minorities.-
Seto M. C. (2017). The Puzzle of Male Chronophilias. Archives of sexual behavior, 46(1), 3–22. https://doi.org/10.1007/s10508-016-0799-y
Line 63 = I do not think authors are able to use the word “correlation” for such analysis was not conducted in the cited literature. I suggest to use instead “overlap”
Line 67 = authors seem to only use the value for clinical settings to justify their point. They may want to expand the argument’s repertoire.
Line 117 = Why using Identity-Behavior discordance (hence IBD), and not one that also includes Attraction as the model suggests?
Line 127 = I believe ref 21 does not covered British people.
Lines 128-132 = these statements need, probably, more than one reference.
Lines 133-138 = this statement is problematic. There are several studies exploring sexual fluidity in heterosexual people, just as much as heterosexual discordance. One cannot say that a researcher may start from zero in the exact question researchers state in this short paragraph. Importantly, whereas it is one way or the other, it is clear that fluidity may lead to discordance, the other way around, or that both processes/phenomena co-occur.
Paragraphs between lines 154 and 177 should be together/one.
Line 178 = the objective comes out of nowhere. Researchers do a good job to flow from one topic to another. Yet, I believe they are missing a problematization, that ultimately to their Objective. Related to thin, line 189 to 193 appears to be part of the Methods than the Objective.
Methods
Researchers say they classified people into sexes, but use gender categories to describe the two groups, or switch back to males and females. This needs to be straighten up.
Line 248 = it would help to say how many were not sexually active
Line 252 = I think saying “sexuality” when talking about only behaviors is too much of a concept to describe “adultery, one-night stands, and sex without love.” Authors ought to find a better describer, as well as using “i.e.,” if using parentheses.
Line 271 = why is the # of sexual partners conceptualized as a risk behavior? I understand that the *perceived risk of STI is, but the # of sexual partners in itself, inheritably, cannot be considered a sexual risk behavior.
Line 274-275 = If researchers dichotomized the four categories, they need to make clear if they divided the four alternatives into two categories of two, or differently. Also, I the number 1 should be within the parenthesis?
The 2.3. Procedure section should cite the manuscript of the original study methods.
Line 332 = authors speak about conducting logistic, lineal, and multinomial regressions, when they just use Odds ratio’s. This is unnecessary and more on the wrong side than right. Furthermore, authors do not explain for which variables they will calculate Odds ratio’s. Finally, they do not explain as to why the classify age into those age groups.
Results
Table 1 = Any reason as to why not to give the full detail on ethnicity, as opposed to just white an “others”? Same with “other” religions. Because I do not see any. Also, it is confusing that % is presented in the columns as within the parentheses, and in the table they are not. Finally “a” is not defined.
P values in such large samples require effect sizes. Otherwise, they can be highly misleading, much like with the differences reported in Table 2. Still, whereas effect sizes may be considered unnecessary, I would argue the same for statistical comparisons of groups sizes.
Line 445 = it is quite convenient to report “significant” differences and they say that the critical level was different. Did the authors also find other significant differences between 0.051 and 0.1? we would never know.
Figure 1 has no Y axis label. Also, the figure appears not to have a top grey line like the other borders.
Discussion
Findings on IBD are discussed against two studies. Whereas results are descriptive and to some degree inferential, authors are required to say a bit more than just a reiteration than what they found, especially against more literature.
Authors mention that there are “a number” of limitations, yet mentioned a few. The first is not a limitation. Every study could have done more or differently. Limitations are mainly focused towards what was done, not what could have, especially when there are “a number” of them. Subsequently, there are indeed some groups with small sample sizes (e.g., 37, 44, 66, 77, etc.). But, authors know too these people represent a fraction of the population. They managed to analyze a representative sample of a large country, and even then they had a small sample in these groups for these attributes do not conglomerate many people in the population. Then, I do not see the “small sample” as a limitation, but as a merit of even being able to detect these groups of people. If they believe they can get more, they may be dreaming a bit too big. Finally, and similarly, stating they cannot make causal inferences in an almost purely descriptive study is also not a limitation. If I conduct a study to evaluate depression with psychometric scales I cannot say that I could have used other techniques such as brain imaging to establish causal inference because it was never the intent to do so. A limitation would be to not have had a clinical diagnosis for depression. Yet, neither the first nor the last limitations really count, and the second lacks of practical perspective. Therefore, this section needs to be re-evaluated in its entirety.
Conclusions
Authors do explain what they mean with their suggestion on the (clinical) interventions, where clinicians should be aware that sexual orientation goes far beyond identity. However, they do nothing to develop or discuss (in the discussion) anything on “policy or programs to address disparities and improve outcomes for discordant heterosexual and non-heterosexual populations”.
Comments on the Quality of English Language
These are the least of my corrections. They can be found in my other comments.
Author Response
Please find attached our response to your comments. Please note that in the manuscript you will find the changes suggested by reviewer 1 and by you (reviewer 2).

Round 2
Reviewer 2 Report
Comments and Suggestions for Authors
The authors addressed issues and comments in a satisfactory manner. I have no further significant comments to make.